# On a Gradient Approach to Chebyshev Center Problems with Applications to Function Learning

**Abhinav Raghuvanshi**                                     *200040008@iitb.ac.in*
*Indian Institute of Technology, Bombay*

**Mayank Baranwal**                                         *mbaranwal@iitb.ac.in*
*Tata Consultancy Services Research, Mumbai*
*Indian Institute of Technology, Bombay*

**Debasish Chatterjee**                                     *dchatter@iitb.ac.in*
*Indian Institute of Technology, Bombay*

**Reviewed on OpenReview:** *https://openreview.net/forum?id=lPZVsDhyj3*

## Abstract

We introduce `gradOL`, the first gradient-based optimization framework for solving Chebyshev center problems, a fundamental challenge in optimal function learning and geometric optimization. `gradOL` hinges on reformulating the semi-infinite problem as a finitary max-min optimization, making it amenable to gradient-based techniques. By leveraging automatic differentiation for precise numerical gradient computation, `gradOL` ensures numerical stability and scalability, making it suitable for large-scale settings. Under strong convexity of the ambient norm, `gradOL` provably recovers optimal Chebyshev centers while directly computing the associated radius. This addresses a key bottleneck in constructing stable optimal interpolants. Empirically, `gradOL` achieves significant improvements in accuracy and efficiency on 34 benchmark Chebyshev center problems from a benchmark `CSIP` library. Moreover, we extend `gradOL` to general convex semi-infinite programming (CSIP), attaining up to 4000× speedups over the state-of-the-art `SIPAMPL` solver tested on the indicated `CSIP` library containing 67 benchmark problems. Furthermore, we provide the first theoretical foundation for applying gradient-based methods to Chebyshev center problems, bridging rigorous analysis with practical algorithms. `gradOL` thus offers a unified solution framework for Chebyshev centers and broader CSIPs.

## 1 Introduction and Problem Formulation

Optimal interpolation and learning have been long-standing open problems in signal processing and the theory of function learning. The problem was posed in Micchelli & Rivlin (1977) in the context of signal processing, and while tractable solutions were sought by a multitude of researchers over the subsequent decades, satisfactory solutions continued to remain elusive. In the language of approximation theory, the optimal interpolation (learning) corresponds to the so-called ***Chebyshev center*** problem, which involves constructing a single function from a given class that best justifies a given data set. Beyond function learning, the Chebyshev center problem has found extensive applications in diverse fields, such as robust optimization for managing uncertainty Ben-Tal & Nemirovski (1998), sensor network localization for estimating node positions Doherty & Ghaoui (2001), and facility location problems to optimize placement strategies Drezner & Hamacher (2004). Moreover, its utility extends to tasks like data fitting in classification Tax & Duin (1999), control system design for ensuring stability under bounded uncertainties Zhou et al. (1996), and parameter estimation in error-bounded scenarios Boyd & Vanderberghe (2004), underscoring its fundamental importance in both theoretical and practical domains.

Let us briefly recall the optimal interpolation problem in the context of function learning: At its core, on a Banach space $(\mathfrak{X}, \|\cdot\|)$, one is given a closed and bounded subset $K \subset \mathfrak{X}$, and charged with the task of determining a ball of smallest radius containing $K$. The center $\zeta$ of such a ball is a **_Chebyshev center_** of $K$. In other words, a Chebyshev center of $K$ is an optimizer of the variational problem

$$\underset{f \in \mathfrak{X}}{\text{minimize}} \quad \sup_{g \in K} \|f - g\|. \tag{1}$$

Depending on certain properties (e.g., strict convexity) of the norm $\|\cdot\|$, a set $K$ may not have a single Chebyshev center. The optimal value of (1) is the **_Chebyshev radius_** $r_K$ of $K$. We refer the reader to the review Alimov & Tsar'kov (2019) and the authoritative text (Alimov & Tsar'kov, 2021, Chapter 16) for background on the Chebyshev center problem, and the recent work Binev et al. (2024) for an historical account of optimal interpolation interpreted from the viewpoint of the Chebyshev center. While the problem is posed above in its most general, infinite-dimensional form, any computational method must necessarily operate on a finite-dimensional version (Alimov & Tsar'kov, 2021, Section 16.1). This paper is dedicated to solving this important finite-dimensional problem, for which we present an algorithm to find a solution that is exact up to the precision of the computing machine.

### Function learning and the Chebyshev center

In the context of function learning and interpolation theory, the Chebyshev center problem encodes the idea of *optimal learning* in a hypothesized model class: Here $\mathfrak{X}$ stands for the space of functions whose subsets are the hypothesis classes, and the set $K$ represents the subset of the model class of functions satisfying a given data set. The corresponding optimization problem (1) faces difficult numerical challenges because first, for each fixed $f \in \mathfrak{X}$, the inner maximization over $g$ in (1) is on a potentially infinite-dimensional subset of the Banach space $\mathfrak{X}$, and its solutions cannot be parametrically expressed in closed form in $f$, and second, the outer minimization over $f$ in (1) is also over an infinite-dimensional Banach space $\mathfrak{X}$ in general. Consequently, (1) is numerically intractable. To ensure computational tractability, one "discretizes" the various infinite-dimensional objects in (1) above, and works in a finite dimensional setting;[1] the resulting mathematical optimization is an instance of the so-called *relative Chebyshev center problem*: A **_relative Chebyshev center_** of a closed and bounded subset $K$ with respect to a nonempty $X \subset \mathfrak{X}$ is given by an optimizer of

$$\underset{f \in X}{\text{minimize}} \quad \sup_{g \in K} \|f - g\|, \tag{2}$$

where the set $X$ is chosen to be a reasonably fine *finite* discretization approximating the Banach space $\mathfrak{X}$ and the model class $K$ is restricted to a suitable finite dimensional set of functions (nearly) satisfying the given data. Consequently, it addresses the optimal function learning problem via optimal interpolation from finite datasets. Nevertheless, even the resulting simplified problem (2) continues to be numerically challenging since even in finite-dimensions, the complexity increases exponentially in the state dimension (Alimov & Tsar'kov, 2021, Chapter 15, p. 362). Indeed, the (finite-dimensional) Chebyshev center problem is known to be **_NP-hard_** (Alimov & Tsar'kov, 2021, Chapter 15) in general.

### Chebyshev center via convex semi-infinite programs

Some NP-hard problems may permit numerically viable and robust approximations, and it turns out that such is the case with the (finite dimensional) Chebyshev center problem. *At its core*, given a nonempty and compact set $K \subset \mathbb{R}^n$, the centerpiece of a numerical algorithm must operate to solve the finite-dimensional variational problem

$$r_K = \inf_{x \in X} \sup_{u \in K} \|u - x\|, \tag{3}$$

where $X$ is either $\mathbb{R}^n$ or a nonempty convex subset thereof. Function learning from finite data can be framed in the language of (3) (see Paruchuri & Chatterjee (2023)). Note that (3) is *always* feasible, and if the set

---

[1] For computational tractability, as pointed out in (Alimov & Tsar'kov, 2021, Section 16.1), it is imperative to restrict attention to finitary objects; consequently, considering finite-dimensional avatars of the various objects in (1) is the best that can be done.

$K$ contains more than two distinct points, then the Chebyshev radius of $K$ is non-zero. Introducing a slack variable $s \in \mathbb{R}$, we note that the value $r_K$ of (3) is precisely equal to the value of

$$
\begin{aligned}
&\underset{(s,x)}{\text{minimize}} && s \\
&\text{subject to} && \begin{cases} \|u - x\| - s \leqslant 0 & \text{for all } u \in K, \\ (s, x) \in \mathbb{R} \times X, \end{cases}
\end{aligned}
\tag{4}
$$

which is a convex semi-infinite program (CSIP) — a finite-dimensional convex optimization problem with a compact (and infinite) family of convex inequality constraints. It follows that solving (4) gives a solution to (3) in the sense that if $(s^*, x^*)$ solves (4), then $r_K = s^*$ and $x^*$ is a Chebyshev center of $K$. The recent work Das et al. (2022) established a viable approach to obtaining exact solutions to CSIPs, and Paruchuri & Chatterjee (2023) developed the connection between optimal function learning and CSIPs to provide a numerically viable solution to the Chebyshev center problem. While these two works established general principles and high-level algorithms, robust numerical algorithms for solving the Chebyshev center problem were outside their scope. This article is devoted to the development of *a gradient based algorithm for solving the Chebyshev center problem*, which is enabled by reformulating the underlying semi-infinite program into an entirely finitary max-min structure, which in turn *serves as a numerically viable algorithm for optimal function learning*

### Why Are Chebyshev Center Problems Practically Important? - Illustrative Examples

Chebyshev center problems, at their core, capture the principle of constructing an "optimally central" object within a set $K$, which translates directly into optimal function learning when the set represents candidate hypotheses consistent with observed data. Beyond their foundational role in approximation theory, several concrete examples illustrate their broad impact:

**Robust Control Systems**: In designing a controller for a physical system (e.g., an aircraft or a robot), the exact parameters of the system are often uncertain due to manufacturing tolerances or environmental changes. The set $K$ can represent the set of all possible system responses. The Chebyshev center corresponds to a single controller design that guarantees the best possible performance (e.g., stability) in the worst-case scenario, across all possible system variations in $K$.

**Sensor Network Localization**: Consider estimating the position of a sensor node based on signals from several fixed beacons. Each signal constrains the node's location to be within a certain region. The intersection of these regions forms the set $K$ of all possible locations. The Chebyshev center of $K$ provides the optimal location estimate, minimizing the maximum possible error between the estimate and true location.

**Financial Portfolio Optimization**: An investor might model a set $K$ of plausible future market scenarios, each represented by a vector of asset returns. The goal is to construct a single investment portfolio (the center) that minimizes the maximum regret across all scenarios in $K$. This approach creates a portfolio optimally hedged against worst-case market outcome.

### Contributions

Chebyshev center problems and more broadly, CSIPs, are notoriously challenging to solve, even in moderately high-dimensional settings. The core difficulty lies in their semi-infinite nature: the feasible region is defined by infinitely many constraints, rendering even approximate solutions computationally demanding. In response, prior work has often focused on tractable relaxations, such as the *relaxed Chebyshev center (RCC)* problem Eldar et al. (2007); Xia et al. (2021), which approximates the original formulation.

Consequently, developing a scalable, gradient-based solver for the **original (non-relaxed) Chebyshev center problem has remained a longstanding open challenge** in optimization and function learning. Despite the importance of the problem, there has been little progress on algorithms that can directly tackle the original formulation using modern optimization toolkits. This paper introduces `gradOL`, **the first framework to successfully address this gap**. To the best of our knowledge, no existing methods have successfully exploited gradient-based optimization to directly solve the Chebyshev center problem, a task for which we leverage modern automatic differentiation techniques. Relying on the fundamental insights of Borwein (1981), the key results of Paruchuri & Chatterjee (2023) can be leveraged to establish that the value

of (4) is precisely equal to the value of

$$\sup_{(u_1,\ldots,u_{n+1})\in K^{n+1}} \inf_{(s,x)\in\mathbb{R}\times X}\left\{s \,\middle|\, \|u_i - x\| \leqslant s,\ i = 1,\ldots,n+1\right\}. \tag{5}$$

Observe that (5) is entirely *finitary*: The inner minimization is a standard convex optimization problem with $n+1$ constraints, while the outer maximization is global and over $n+1$ many variables, each of which is $n$-dimensional. Since there is no realistic possibility of obtaining an expression of the map

$$K^{n+1} \ni (u_1,\ldots,u_{n+1}) \mapsto \mathcal{G}(u_1,\ldots,u_{n+1})$$

where, $\mathcal{G}(u_1,\ldots,u_{n+1})$ is defined as:

$$\inf_{(s,x)\in\mathbb{R}\times X}\left\{s \,\middle|\, \|u_i - x\| \leqslant s \text{ for } i = 1,\ldots,n+1\right\} \in \mathbb{R},$$

but it can be evaluated at will by employing standard convex optimization solvers, zeroth-order numerical methods are natural candidates for solving (5). However, the availability of autodiff libraries, e.g., `pytorch` and `zygote`, raises the pertinent question of whether employing first-order (sub-)gradient methods (leveraging automatic differentiation) to solve the outer maximization in (5) would be feasible. Below we summarize our primary contributions:

1. **Efficient solver for Chebyshev center problems**: We present a robust and numerically efficient implementation of a gradient-based optimization technique for the Chebyshev center problem. Our complete `Julia` package, `gradOL`, employs the latest automatic differentiation libraries to compute (sub-)gradients, ensuring high accuracy and scalability. Specifically:
   - The optimal value obtained via our solver corresponds directly to the Chebyshev radius.
   - When the ambient norm on $\mathbb{R}^n$ is strongly convex, our algorithm also produces optimizers that attain the Chebyshev radius through the formulation in (5). This is particularly valuable in optimal function learning, where the construction of numerically viable optimal interpolants has remained a notable challenge over decades.

2. **Theoretical foundation for gradient-based learning**: The inner minimization in (5) is subject to constraints, raising two key challenges: (a) whether the gradient of the inner objective, as required by the outer maximization step, is well-defined, and (b) how to compute this gradient efficiently when it exists. While our proposed `gradOL` algorithm directly addresses the computational aspect in (b) through automatic differentiation, we also provide a rigorous theoretical foundation that justifies the use of gradient-based learning in the first place. Specifically, we establish that the map $u \mapsto \mathcal{G}(u)$ is locally Lipschitz, thereby ensuring the existence of generalized gradients.

3. **Benchmark testing and performance**: The `gradOL` package has been tested extensively on a curated collection of 34 benchmark Chebyshev center problems from the CSIP library Vaz (2001). As reported in the subsequent sections, `gradOL` demonstrates consistent improvements over existing techniques, providing very accurate and ***vastly more efficient*** solutions in all cases.

4. **The case of general CSIPs**: Since the Chebyshev center problem is a special case of CSIPs, our gradient-based approach naturally extends to solving general CSIPs. We applied `gradOL` to an extended library of 33 additional benchmark CSIPs. Notably, `gradOL` shows ***several orders of magnitude improvement in speed*** in almost all cases, and the values reported by `gradOL` are no worse than the best reported values in all but two cases.

## 2 Preliminaries

For Chebyshev center problems, both geometric and optimization-based methods have been proposed. In the case of certain convex sets, the Chebyshev center problem can be formulated as a linear program (LP) when the feasible region is defined by linear inequalities, making it computationally efficient to solve. Recent developments have focused on numerical algorithms that combine targeted sampling techniques and convex semi-infinite optimization to compute the Chebyshev radius and center more efficiently. These methods are particularly relevant in optimal learning scenarios, especially when working with compact hypothesis spaces within Banach spaces (Paruchuri & Chatterjee, 2023).

**A key result from the theory of CSIPs**

A CSIP is a finite-dimensional convex optimization problem with infinitely many constraints. Consider

$$\begin{aligned}
&\underset{y \in \mathcal{Y}}{\text{minimize}} && f_\circ(y) \\
&\text{subject to} && \begin{cases} f(y,v) \leqslant 0 & \text{for all } v \in \mathcal{U}, \\ \mathcal{Y} \subset \mathbb{R}^d, \mathcal{U} \subset \mathbb{R}^m, \end{cases}
\end{aligned} \tag{6}$$

along with the following data:

((6)-a) $\Omega \subset \mathbb{R}^d$ is an open convex set, $\mathcal{Y} \subset \Omega$ is a closed and convex set with non-empty interior, and $\mathcal{U}$ is a compact set,

((6)-b) $f_\circ : \Omega \to \mathbb{R}$ is a continuous convex function,

((6)-c) $f : \Omega \times \mathcal{U} \to \mathbb{R}$ is a continuous function such that $f(\cdot, v)$ is convex for every $v \in \mathcal{U}$, and

((6)-d) the admissible set $\bigcap_{v \in \mathcal{U}} \{y \in \mathcal{Y} \mid f(y,v) \leqslant 0\}$ has non-empty interior.

The following recent result (Das et al., 2022, Theorem 1, Proposition 2) provides a numerically viable mechanism to solve CSIPs with low memory requirements, and constitutes the backbone of the gradient technique established in this article.

**Theorem 2.1.** *Consider the CSIP* (6) *along with its associated data* ((6)-a) – ((6)-d). *Define the function*

$$\mathcal{U}^d \ni (v_1, \ldots, v_d) =: \bar{v} \mapsto \mathcal{G}(\bar{v}) := \inf_{y \in \mathcal{Y}} \left\{ f_\circ(y) \mid f(y, v_i) \leqslant 0 \text{ for } i = 1, \ldots, d \right\}. \tag{7}$$

*Then the (optimal) value of the CSIP* (6) *equals*

$$\sup_{(v_1, \ldots, v_d) \in \mathcal{U}^d} \mathcal{G}(v_1, \ldots, v_d). \tag{8}$$

*Moreover, if* (6) *admits a unique solution and if* $(v_1^\circ, \ldots, v_d^\circ)$ *maximizes* $\mathcal{G}$, *then an optimizer of*

$$\inf_{y \in \mathcal{Y}} \left\{ f_\circ(y) \mid f(y, v_i^\circ) \leqslant 0 \text{ for } i = 1, \ldots, d \right\}$$

*also optimizes* (6).

Theorem 2.1 is applicable the CSIP (4) corresponding to the Chebyshev center problem (3); indeed, one picks $d = n+1$, $m = n$, $\Omega = \mathbb{R} \times \mathbb{R}^n$, $\mathcal{U} = K \subset \mathbb{R}^n$, $\mathcal{Y} \subset \Omega$ is some closed and convex set containing $[0, +\infty[ \times K$, and the functions $\Omega \ni (s,x) \mapsto f_\circ(s,x) = s$ and $\Omega \times K \ni ((s,x), u) \mapsto f((s,x), u) = \|u - x\| - s$ in (6) to obtain (4). It is well-known that if the underlying norm $\|\cdot\|$ in (4) is strictly convex, e.g., if it is the Euclidean norm, then the Chebyshev center is unique. Consequently, the CSIP (4) admits a unique solution, and in the light of Theorem 2.1, the value of (6) equals that of (5) (as noted before (5)) and an optimizer of the Chebyshev center problem may be extracted as indicated in Theorem 2.1; of course, the corresponding (optimal) value is the Chebyshev radius.

In the case of the CSIP (4), the function $\mathcal{G}$ becomes

$$K^{n+1} \ni (u_1, \ldots, u_{n+1}) =: \bar{u} \mapsto \mathcal{G}(\bar{u}) := \inf_{(s,x) \in \mathbb{R} \times X} \left\{ s \mid \|u_i - x\| - s \leqslant 0 \text{ for } i = 1, \ldots, n+1 \right\}. \tag{9}$$

Our main contribution — the algorithm `gradOL` — leverages insights from Theorem 2.1 to develop a (sub-)gradient-based optimization algorithm, `gradOL`, which employs automatic differentiation for numerical gradient estimation. `gradOL` is designed to employ numerical gradients to maximize $\mathcal{G}$. A key challenge in maximizing $(u_1, \ldots, u_{n+1}) \mapsto \mathcal{G}(u_1, \ldots, u_{n+1})$ lies in the absence of closed-form analytical formulae for $\mathcal{G}$ and therefore its gradients. The success of gradient algorithms for optimization hinges on the local Lipschitz property of the corresponding objective function, so it is natural to find conditions under which this property holds for $\mathcal{G}$ in (9).

**Remark 2.2** (Convexity in the inner variable vs. outer maximization)**.** Although the original problem is convex, it is semi-infinite. Theorem 2.1 yields a finitary max–min reformulation with the same optimal value. For any fixed $\overline{u}$, the inner minimization remains a convex program; in contrast, the outer maximization over $\overline{u}$ is generally non-concave and can be non-smooth. This is the source of the stationarity-type guarantees discussed in Remark 3.2.

### Automatic Differentiation for Gradient Computation

Our method leverages automatic differentiation (AD) (Baydin et al., 2018), which computes exact gradients by propagating derivatives through the computational graph of the routine. The value function $\mathscr{G}$ in (7) is defined implicitly through an inner optimization, and its gradient has no closed form. AD differentiates through the full sequence of operations that produce $\mathscr{G}$, yielding an exact $\nabla\mathscr{G}$ and enabling the `gradOL` algorithm.

## 3 A Differentiable Max-Min Formulation and the `gradOL` Algorithm

Ensuring local Lipschitzness of the function $\mathscr{G}$ for general CSIPs with inequality constraints presents significant difficulties, but certain structures of Chebyshev center problems make them amenable to apply results from the shelf for proving local Lipschitzness of the corresponding $\mathscr{G}$. Proposition A.1 in Appendix A provides a set of sufficient conditions for $\mathscr{G}(\cdot)$ to be continuous. In order to satisfy the hypotheses of Proposition A.1, it is possible to rephrase the Chebyshev center problem (3) to the smooth variant

$$\inf_{x \in X} \sup_{u \in K} \|u - x\|^2, \tag{10}$$

such that its optimal value is the square of the Chebyshev radius and its corresponding CSIP (4) features the twice continuously differentiable objective function $f_\circ(s, x) := s$ and constraint function $f((s, x), u) = \|u - x\|^2 - s$.

Consider the CSIP corresponding to (10) given by

$$\inf_{(s,x) \in \mathbb{R} \times X} \left\{ s \mid \|u - x\|^2 - s \leqslant 0 \quad \text{for all } u \in K \right\}, \tag{11}$$

for which we have the following result. Recall that $K$ may be replaced by its closed convex hull, $\overline{\mathrm{conv}}(K)$, in the Chebyshev center problem; consequently, we shall assume in the sequel that $K = \overline{\mathrm{conv}}(K)$ without losing generality.

**Theorem 3.1.** *Consider the problem* (11) *with $K \subset \mathbb{R}^n$ non-empty and compact,[2] and suppose that $X$ is a compact and convex set containing $K$. Define the function*

$$K^{n+1} \ni (u_1, \ldots, u_{n+1}) =: \overline{u} \mapsto \mathscr{G}(\overline{u}) := \inf_{(s,x) \in \mathbb{R} \times X} \left\{ s \mid \|u_i - x\|^2 \leqslant s \text{ for } i = 1, \ldots, n+1 \right\}. \tag{12}$$

*Then the mapping $\mathscr{G}$ is $\mathbb{R}$-valued, Lipschitz continuous, and the value of* (10) *is precisely equal to* $\sup_{\overline{u} \in K^{n+1}} \mathscr{G}(\overline{u})$.

*Proof.* The mapping $\mathscr{G}$ is $\mathbb{R}$-valued since the admissible set is non-empty. Indeed, since $K$ is non-empty and compact, it is bounded. Consequently, $\mathscr{G}$ admits an upper bound of $\operatorname{diam} K := \sup_{x', x'' \in K} \|x' - x''\|$, while $0$ is an obvious lower bound of $\mathscr{G}$. To wit, $\mathscr{G}$ is $\mathbb{R}$-valued.

For $s' > \operatorname{diam}(K)$ and arbitrary $x' \in K$ we have $\|x - u\|^2 - s' < 0$ for all $u \in K$, which means that (11) is strictly feasible. Therefore, the strict feasibility condition ((6)-d) in the context of (6) holds in the case of (11). The remaining hypotheses of Theorem 2.1 are clearly satisfied for (11), and its assertion guarantees that the value of (10) is $\sup_{\overline{u} \in K^{n+1}} \mathscr{G}(\overline{u})$.

It remains to prove Lipschitz continuity of $\mathscr{G}$, which we present in a sequence of steps:

---

[2]Recall that by assumption $K$ is closed and convex.

**Step 1**: The optimization problem on the right-hand side of (12) admits a solution for every $\overline{u} \in K^{n+1}$. Indeed, this is an immediate consequence of Weierstrass's theorem in view of continuity of the objective and constraint functions and compactness of the admissible set.

**Step 2**: The set-valued map $S_{\mathrm{opt}} : K^{n+1} \rightrightarrows \mathbb{R}^n$ defined by

$$K^{n+1} \ni \overline{u} \mapsto S_{\mathrm{opt}}(\overline{u}) := \underset{(s,x)\in\mathbb{R}\times X}{\arg\min} \left\{ s \;\middle|\; \|x - u_i\|^2 \leqslant s \text{ for } i = 1,\ldots,n+1 \right\}$$

is a mapping. Indeed, that $S_{\mathrm{opt}}$ is non-empty valued follows from **Step 1**. Moreover, since $K \neq \varnothing$ by hypothesis, for any $y \in K$ and for $\overline{u} = (u_1,\ldots,u_{n+1})$ the pair $\left(\max_{i=1,\ldots,n+1}\|u_i - y\|^2, y\right) \in \mathbb{R} \times X$ is in the admissible set of the indicated optimization problem. Moreover, the objective function of the optimization problem is such that if $(s', x')$ and $(s'', x'')$ are two optimizers, then of course $s' = s''$, which means that two optimizers can differ only in the second component. But then due to strong convexity of the norm $\|\cdot\|$, the constraints for $k = 1,\ldots,n+1$ give us

$$\left\|2^{-1}x' + 2^{-1}x'' - u_k\right\| < 2^{-1}(\|x' - u_k\| + \|x'' - u_k\|) < \sqrt{s'},$$

implying that neither $(s', x')$ nor $(s', x'')$ is an optimizer (because one obtains a better value for the pair $\left(\max_{i=1,\ldots,n+1}\left\|2^{-1}(x' + x'')\right\|^2, 2^{-1}(x'+x'')\right) \in \mathbb{R} \times X$, which is admissible due to convexity of $K$), contradicting our premise. Uniqueness of optimizers follows, and therefore, $S_{\mathrm{opt}}$ is a mapping.

**Step 3**: Fix $\overline{u} \in K^{n+1}$ and consider the convex nonlinear program

$$\inf_{(s,x)\in\mathbb{R}\times X} \left\{ s \;\middle|\; \|u_i - x\|^2 \leqslant s \text{ for } i = 1,\ldots,n+1 \right\} \tag{13}$$

on the right-hand side of (12). The family of $(n+1)$ constraints in the preceding program admits the Jacobian (derivative) matrix $J(\overline{u}; x) := \begin{pmatrix} -1 & 2(x-u_1)^{\mathsf{T}} \\ -1 & 2(x-u_2)^{\mathsf{T}} \\ \vdots & \vdots \\ -1 & 2(x-u_{n+1})^{\mathsf{T}} \end{pmatrix}$ for $x \in K$, and clearly $v := \begin{pmatrix} 1 & 0_{n+1}^{\mathsf{T}} \end{pmatrix}^{\mathsf{T}} \in \mathbb{R}^{n+1}$ satisfies $J(\overline{u}; x) \cdot v = \begin{pmatrix} -1 & \cdots & -1 \end{pmatrix}^{\mathsf{T}}$. Therefore, the Mangasarian-Fromovitz constraint qualification (MFCQ) condition holds for (13) for the entire family of constraints, and consequently, also for the active constraints. In addition, the set-valued map $S_{\mathrm{feas}}$ defined by

$$K^{n+1} \ni \overline{u} \mapsto S_{\mathrm{feas}}(\overline{u}), S_{\mathrm{feas}}(\overline{u}) := \bigcap_{i=1}^{n+1} \left\{ (s,x) \in \mathbb{R} \times X \;\middle|\; \|u_i - x\|^2 - s \leqslant 0 \right\} \subset \mathbb{R}^n \tag{14}$$

is uniformly compact because its image is contained in the compact set $[0, \operatorname{diam} K] \times X \subset \mathbb{R} \times X$ due to the definition of (13). (Fiacco & Ishizuka, 1990, Theorem 4.2) — reproduced in Proposition A.1 in Appendix §A, therefore, applies to (13), and guarantees local Lipschitz continuity of $\mathscr{G}$ around $\overline{u}$.

**Step 4**: The map $\mathscr{G}$ is Lipschitz continuous. This follows from (Cobzaş et al., 2019, Theorem 2.1.6), but we provide a quick proof here. Since $\overline{u}$ was selected in **Step 3** arbitrarily, we get the local Lipschitz property of $\mathscr{G}$ around every $\overline{u} \in K^{n+1}$, and consequently, $\mathscr{G}$ is continuous. Since $K^{n+1}$ is compact, $\mathscr{G}$ attains its maximum and minimum on $K^{n+1}$, and let $M := \sup_{\overline{u}\in K^{n+1}} \mathscr{G}(\overline{u}) - \inf_{\overline{u}'\in K^{n+1}} \mathscr{G}(\overline{u}')$. If $\mathscr{G}$ is *not* Lipschitz on $K^{n+1}$, then $\sup_{\substack{\overline{u},\overline{u}'\in K^{n+1} \\ \overline{u}\neq\overline{u}'}} \frac{|\mathscr{G}(\overline{u}) - \mathscr{G}(\overline{u}')|}{\|\overline{u}-\overline{u}'\|} = +\infty$. That means there exist sequences $(\overline{u}_k)_{k\in\mathbb{N}^*}, (\overline{u}'_k)_{k\in\mathbb{N}^*} \subset K^{n+1}$ such that $\lim_{k\to+\infty} \frac{|\mathscr{G}(\overline{u}_k) - \mathscr{G}(\overline{u}'_k)|}{\|\overline{u}_k-\overline{u}'_k\|} = +\infty$. Since $K^{n+1}$ is compact, it follows that there exist subsequences $(\overline{u}_{k_\ell})_{\ell\in\mathbb{N}^*}$ and $(\overline{u}'_{k_\ell})_{\ell\in\mathbb{N}^*}$ that converge to some $\overline{v}$ and $\overline{v}'$ in $K^{n+1}$, respectively. But then, since $\mathscr{G}$ is bounded on $K^{n+1}$, the numerator of the preceding limit is bounded by $2M$, which gives us

$$\left(\left\|\overline{u}_{k_\ell} - \overline{u}'_{k_\ell}\right\| \xrightarrow[\ell\to+\infty]{} 0\right) \quad \Rightarrow \quad (\overline{v} = \overline{v}').$$

But this leads to $\lim_{\ell\to+\infty} \frac{|\mathscr{G}(\overline{u}_{k_\ell}) - \mathscr{G}(\overline{v}')|}{\|\overline{u}_{k_\ell}-\overline{v}'\|} = +\infty$, contradicting local Lipschitz continuity of $\mathscr{G}$ around $\overline{v}'$, and thus asserting Lipschitz continuity of $\mathscr{G}$ on $K^{n+1}$. □

Theorem 3.1 relies on the set $K$ being a compact subset of $\mathbb{R}^n$. While $K$ could be discrete for this result to hold, the application of gradient based techniques such as `gradOL` is reliant on the set $K^{n+1}$ being a reasonably "nice" (e.g., convex) subset of $\mathbb{R}^m$, which is true in a majority of applications.

**The algorithm `gradOL`**

Our chief contribution, the algorithm `gradOL` described below, operates within an iterative framework, constructing $\mathcal{G}(u_1, \ldots, u_{n+1})$ as a computational graph rooted at the variable nodes $(u_1, \ldots, u_{n+1})$. For each given $\bar{u} = (u_1, \ldots, u_{n+1})$, the inner constrained minimization over $(s, x)$ is solved using an unconstrained log-barrier method. This process iteratively updates $(s_k, x_k)$, starting from an initial guess $(s_0, x_0)$, while preserving the computational graph dependency on $\bar{u}$, thereby explicitly representing $(s, x)$ as a function of $\bar{u}$. Once the inner optimization reaches optimality, $\mathcal{G}(\bar{u})$ can be computed, and its numerical gradients are obtained via backpropagation through the computational graph, eliminating the need for explicit gradient derivation. We now present the `gradOL` algorithm.

Let $\alpha > 0$ be a "barrier parameter" (Boyd & Vanderberghe, 2004), and let $\ell(\cdot)$ be a (safe) log function, e.g. $\ell(t) := \log(\max(t, \epsilon))$ for a small $\epsilon > 0$. Given a current estimate $\bar{u}_k := (u_1, \ldots, u_{n+1}) \in K^{n+1}$ at iteration $k$, we solve the inner problem

$$\min_{(s,x) \in \mathbb{R} \times X} \left( \alpha s - \sum_{i=1}^{n+1} \ell\left(s - \|x - u_i\|^2\right) \right) := \mathcal{G}(\bar{u}_k). \tag{15}$$

Let $(s_k, x_k)$ be a minimizer of (15). Next, for the outer step, we take a step from $\bar{u}_k$ in the direction that increases a target function $\mathcal{G}(\bar{u}_k)$. In our illustrative gradient-based scheme, we compute $\nabla_{\bar{u}} \mathcal{G}(\bar{u}_k)$, and perform an update to obtain $\bar{u}_{k+1}$. The step size (learning rate) may be constant or adaptive. Note that a reasonable initial estimate for $(s, x)$ can be obtained using standard off-the-shelf convex optimization solvers like CVX or Convex.jl. In our framework, we utilize them to reduce the number of iterative updates required. The use of automatic differentiation libraries (e.g. Zygote in Julia or Autograd in PyTorch) facilitates gradient estimation, while off-the-shelf solvers enhance computational speed via warm starts. For a general CSIP, the objective in (15) is replaced with $f_\circ(x)$, while the log-barrier is used to enforce the constraints $\{f(x, u_i)\}$.

---

**Algorithm 1 `gradOL`**

---

1: **Input:** Initial guess $\bar{u}_0 := (u_1, \ldots, u_{n+1})$; barrier parameter $\alpha > 0$; max epochs M; tolerance $\delta > 0$.
2: **for** $k = 0, 1, 2, \ldots, \text{M-1}$ **do**
3:     Declare $(u_1, \ldots, u_{n+1})$ as variable nodes in computational graph; Initial guess $(s_0, x_0)$
4:     **(Inner) minimization step:**

$$\left(s^*_{k+1}(\bar{u}_k), x^*_{k+1}(\bar{u}_k)\right) \in \underset{(s,x) \in \mathbb{R} \times \mathbb{R}^n}{\arg\min} \left\{ \alpha s - \sum_{i=1}^{n+1} \ell\left(s - \|x - u_i\|^2\right) \right\}$$

    $\left(s^*_{k+1}(\bar{u}_k), x^*_{k+1}(\bar{u}_k)\right)$ is obtained iteratively using off-the-shelf solvers, such as Convex.jl, while retaining the computational graph
5:     **(Outer) Update of $\bar{u}_k$:**
    Define the objective function in terms of $\bar{u}_k$,
$$\mathcal{G}(\bar{u}_k) = f_\circ\left(s^*_{k+1}(\bar{u}_k), x^*_{k+1}(\bar{u}_k)\right)$$

    Compute the gradient $\nabla_{\bar{u}_k} \mathcal{G}(\bar{u}_k)$ using the automatic differentiation library.

    Update $\bar{u}_{k+1} = \bar{u}_k + \eta \nabla_{\bar{u}} \mathcal{G}(\bar{u}_k)$, possibly applying clipping/projection to keep $\bar{u}_{k+1}$ in $K^{n+1}$.
6:     **Check convergence:** If $\|\bar{u}_{k+1} - \bar{u}_k\| < \delta$, then **stop**.
7: **end for**
8: **Return:** $\left(s^*_{\text{M}}(\bar{u}_{\text{M-1}}), x^*_{\text{M}}(\bar{u}_{\text{M-1}})\right)$

---

**Remark 3.2** (On the computational complexity of `gradOL`). The function $\mathcal{G}$ is potentially non-smooth, non-concave, and subject to constraints, which makes deriving convergence guarantees for `gradOL` highly nontrivial and outside the main scope of this work. Nonetheless, by suitably adapting the proof techniques developed in Liu et al. (2024), our preliminary analysis shows that `gradOL` converges to a generalized Gold-

Table 1: `gradOL` performance comparison on Chebyshev Center and CSIP benchmarks

| Method | Chebyshev Center (34 instances) | | CSIP (33 instances) | |
| --- | --- | --- | --- | --- |
| | Solved | Avg. Runtime (ms) | Solved | Avg. Runtime (ms) |
| Iterative Sampling | 9 | 3,880.95 | 20 | 1,695.37 |
| MSA–Simulated Annealing | 19 | 23,768.12 | 20 | 16,417.43 |
| `SIPAMPL` | 32 | 50,906.00 | 28 | 72,336.00 |
| **`gradOL`** | **34** | **638.79** | **33** | **303.77** |

stein stationary point of $\mathcal{G}(\cdot)$ with complexity

$$\mathcal{O}\left(\frac{4G^2 Bc\sqrt{n+1}\,\Delta}{\delta(\epsilon^2 - 4G^2 - 8e^2)}\right)$$

where $\epsilon$ denotes the allowable error in evaluating $\mathcal{G}$. The constants $G$, $B$, and $c$ capture, respectively, the regularity of the objective function, the size of the feasible set, and the details of the smoothing procedure (see Liu et al. (2024)). Here, $n+1$ is the dimension of the uncertainty variable, and $e$ denotes the error in approximating the gradient $\nabla\mathcal{G}$, obtained through automatic differentiation frameworks such as `zygote`. The term $\Delta$ represents the computational cost of the inner minimization solver (e.g., CLARABEL in our case), which scales as $\mathcal{O}(\sqrt{\alpha}\log(1/\epsilon))$ with the barrier parameter $\alpha$. Importantly, `gradOL` achieves a dependence of only $\sqrt{n+1}$ in the outer maximization step—this is the best-known scaling, as existing methods often incur linear or even exponential dependence on $n$. A sketch of the proof for this preliminary result is deferred to Appendix B due to space constraints.

## 4 Numerical Experiments and Benchmarking

We implement `gradOL` in Julia (version 1.8.1)[3] and benchmark its performance on an Ubuntu 24.04 system. The hardware configuration consists of an AWS t2.large instance with 2 vCPUs, backed by an Intel Xeon processor. The instance has 8 GiB of RAM. Using Julia's BenchmarkTools library, we measure the mean runtime of `gradOL` across $10^3$ iterations for each of the 67 **hard** CSIP problems listed in (Vaz, 2001). All computations run on the CPU without GPU acceleration, ensuring that the reported performance reflects only the algorithm and its implementation.

We benchmark `gradOL` against the industry standard `SIPAMPL` package (Vaz et al., 2004), the best-known solver in the literature for solving CSIPs, presented in Table 2. In addition, `gradOL` is also compared with the recently reported simulated-annealing based approach (Paruchuri & Chatterjee, 2023) (a.k.a. the MSA-Simulated Annealing) and an iterative sampling based approach (details in Supplementary). In our experiments, we set the tolerance $\delta$ in Algorithm 1 between $1 \times 10^{-4}$ and $1 \times 10^{-3}$ and limit the maximum iterations M to $10^3$. The barrier parameter $\alpha$ was chosen to be sufficiently large ($\approx 10^5$). Since the algorithm performs outer maximization by computing $\nabla_{\bar{u}}\mathcal{G}$, the learning rate should ideally depend on $\mathcal{G}$. We provide a systematic hyperparameter analysis of `gradOL` under varying learning rates and barrier parameters, as detailed in Appendix E.

For 65 of the 67 problems, the optimal values produced by `gradOL` deviates by less than $10^{-2}$ from previously reported results. The largest absolute errors appear in *watson10* ($2.753\times10^{-1}$) and *honstedel* ($1.697\times10^{-1}$). In particular, `gradOL` achieves an exact match (within the provided accuracy) for the objective function values in 5 problems. Our results highlight `gradOL`'s efficiency, solving each CSIP problem in milliseconds on average. More specifically, `gradOL` achieves a remarkable improvement in runtimes (upto $\approx \mathbf{4 \times 10^3}$) compared to the `SIPAMPL` solver on these benchmarks. The reduced computation time and consistent benchmark performance highlight the algorithm's scalability and real-world potential. Table 1 compares the performances of `gradOL` with the aforementioned algorithms on the benchmark problem instances. Notably, `gradOL` is signifi-

---

[3]Source code is included in the supplementary.

cantly better than all other algorithms, underscoring the importance of gradient-based solvers for Chebyshev center problems and CSIPs alike.

## 5   Conclusion

This work introduces `gradOL`, a novel algorithm for efficiently solving Chebyshev center problems, packaged within a robust `Julia` implementation. Extensive testing on 34 Chebyshev center problems, and more generally, 67 CSIPs demonstrated its superior accuracy and superb computational efficiency over existing solvers. While `gradOL` proves to be highly effective, certain CSIP instances challenge gradient-based methods, highlighting areas for improvement, especially centering around adaptation of the learning rate to the problems under consideration. Future work will focus on enhancing robustness for such cases, extending the theoretical framework, exploring hybrid optimization approaches, and improving scalability to high-dimensional settings, aiming to establish `gradOL` as a versatile tool for broader optimization tasks.

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

Table 2: Comparison of algorithmic performance with benchmark solutions for CSIPs.

| Problem | Value reported | gradOL value | Time reported (ms) | gradOL time (ms) |
|---|---|---|---|---|
| coopeL Price & Coope (1996) | $3.4310 \times 10^{-1}$ | $3.3631 \times 10^{-1}$ | - | 4.067 |
| coopeM Price & Coope (1996) | $1.0000 \times 10^{0}$ | $1.0000 \times 10^{0}$ | - | 3.052 |
| coopeN Price & Coope (1996) | $0.0000 \times 10^{0}$ | $-6.9580 \times 10^{-3}$ | $1.19 \times 10^{3}$ | 3.209 |
| fang1 Fang & Wu (1994) | $4.7927 \times 10^{-1}$ | $4.7939 \times 10^{-1}$ | $2.173 \times 10^{4}$ | 28.737 |
| fang2 Fang & Wu (1994) | $6.9315 \times 10^{-1}$ | $6.7982 \times 10^{-1}$ | $2.307 \times 10^{4}$ | 65.094 |
| fang3 Fang & Wu (1994) | $1.7185 \times 10^{0}$ | $1.7183 \times 10^{0}$ | $2.128 \times 10^{4}$ | 33.526 |
| ferris1 Ferris & Philpott (1989) [†] | $4.8800 \times 10^{-3}$ | $4.9828 \times 10^{-4}$ | $5.99 \times 10^{3}$ | $2.296 \times 10^{2}$ |
| ferris2 Ferris & Philpott (1989) | $-1.7869 \times 10^{0}$ | $-1.7865 \times 10^{0}$ | $7.46 \times 10^{3}$ | 8.177 |
| goerner4 Goerner (1997) [†] | $5.3324 \times 10^{-2}$ | $2.6209 \times 10^{-2}$ | $8.25 \times 10^{3}$ | $2.658 \times 10^{2}$ |
| goerner5 Goerner (1997) [†] | $2.7275 \times 10^{-2}$ | $2.5906 \times 10^{-2}$ | $2.2 \times 10^{4}$ | $1.665 \times 10^{2}$ |
| goerner6 Goerner (1997) [†] | $1.0770 \times 10^{-3}$ | $5.9995 \times 10^{-5}$ | $4.658 \times 10^{4}$ | 47.247 |
| honstedel Honstede (1979) | $1.2124 \times 10^{0}$ | $1.0424 \times 10^{0}$ | - | 3.984 |
| kortanek1 Kortanek & No (1993) | $3.2212 \times 10^{0}$ | $3.2184 \times 10^{0}$ | $4.802 \times 10^{4}$ | 6.018 |
| kortanek2 Kortanek & No (1993) | $6.8629 \times 10^{-1}$ | $6.8628 \times 10^{-1}$ | $1.11 \times 10^{3}$ | $9.324 \times 10^{2}$ |
| kortanek3 Kortanek & No (1993) [†] | $1.4708 \times 10^{-2}$ | $2.2281 \times 10^{-4}$ | $1.5 \times 10^{3}$ | 95.012 |
| kortanek4 Kortanek & No (1993) [†] | $5.2083 \times 10^{-3}$ | $3.7413 \times 10^{-5}$ | $2.666 \times 10^{4}$ | 1.443 |
| leon1 Leon et al. (2000) [†] | $4.5050 \times 10^{-3}$ | $2.4607 \times 10^{-4}$ | $1.29 \times 10^{3}$ | 7.283 |
| leon2 Leon et al. (2000) [†] | $4.1880 \times 10^{-5}$ | $1.0002 \times 10^{-7}$ | $1.111 \times 10^{4}$ | 12.273 |
| leon3 Leon et al. (2000) [†] | $5.2190 \times 10^{-4}$ | $1.0002 \times 10^{-5}$ | $5.12 \times 10^{3}$ | $4.778 \times 10^{2}$ |
| leon4 Leon et al. (2000) [†] | $2.6028 \times 10^{-3}$ | $1.3479 \times 10^{-4}$ | $1.381 \times 10^{4}$ | $6.175 \times 10^{3}$ |
| leon5 Leon et al. (2000) [†] | $1.4257 \times 10^{-2}$ | $4.8963 \times 10^{-4}$ | $4.722 \times 10^{4}$ | 14.99 |
| leon6 Leon et al. (2000) [†] | $1.5540 \times 10^{-4}$ | $1.1692 \times 10^{-5}$ | $4.12 \times 10^{3}$ | 7.378 |
| leon7 Leon et al. (2000) [†] | $2.0997 \times 10^{-3}$ | $2.5904 \times 10^{-4}$ | $4.37 \times 10^{3}$ | 21.41 |
| leon8 Leon et al. (2000) [†] | $5.4222 \times 10^{-2}$ | $1.0002 \times 10^{-7}$ | $2.139 \times 10^{4}$ | $5.146 \times 10^{3}$ |
| leon9 Leon et al. (2000) [†] | $1.6338 \times 10^{-1}$ | $1.6338 \times 10^{-1}$ | $1.696 \times 10^{4}$ | 8.694 |
| leon10 Leon et al. (2000) [†] | $5.3825 \times 10^{-1}$ | $5.3776 \times 10^{-1}$ | $2.65 \times 10^{3}$ | 19.612 |
| leon11 Leon et al. (2000) [†] | $4.8414 \times 10^{-2}$ | $2.3620 \times 10^{-3}$ | $1.28 \times 10^{3}$ | 15.384 |
| leon13 Leon et al. (2000) [†] | $2.3607 \times 10^{-1}$ | $2.3531 \times 10^{-1}$ | $1.26 \times 10^{3}$ | 3.972 |
| leon14 Leon et al. (2000) | $6.6667 \times 10^{-1}$ | $6.6640 \times 10^{-1}$ | $1.4 \times 10^{3}$ | 5.151 |
| leon15 Leon et al. (2000) | $-6.6667 \times 10^{-1}$ | $-6.6657 \times 10^{-1}$ | $9.5 \times 10^{2}$ | 4.234 |
| leon16 Leon et al. (2000) | $1.7263 \times 10^{0}$ | $1.7187 \times 10^{0}$ | $2.2 \times 10^{2}$ | 3.04 |
| leon17 Leon et al. (2000) | $-2.0000 \times 10^{0}$ | $-1.9998 \times 10^{0}$ | $1.9 \times 10^{2}$ | 3.4 |
| leon18 Leon et al. (2000) | $-1.7500 \times 10^{0}$ [*] | $-1.7000 \times 10^{0}$ | $3.63 \times 10^{3}$ | 3.303 |
| leon19 Leon et al. (2000) | $7.8584 \times 10^{-1}$ [*] | $7.7543 \times 10^{-1}$ | $2.12 \times 10^{3}$ | 4.003 |
| leon20 Leon et al. (2000) | $3.2380 \times 10^{-1}$ | $3.2316 \times 10^{-1}$ | $1.68 \times 10^{3}$ | 11.713 |
| leon21 Leon et al. (2000) | $-9.9661 \times 10^{1}$ | $-9.9670 \times 10^{1}$ | $3.73 \times 10^{3}$ | $4.399 \times 10^{3}$ |
| leon22 Leon et al. (2000) | $-1.0472 \times 10^{1}$ | $-1.0472 \times 10^{1}$ | $5.9 \times 10^{2}$ | $4.211 \times 10^{3}$ |
| leon23 Leon et al. (2000) | $-3.0857 \times 10^{1}$ | $-3.0857 \times 10^{1}$ | $5.1 \times 10^{2}$ | 16.434 |
| leon24 Leon et al. (2000) | $-1.1998 \times 10^{1}$ [*] | $-1.0370 \times 10^{1}$ | $1.54 \times 10^{3}$ | 7.526 |
| lin1 Lin et al. (1998) | $-1.8244 \times 10^{0}$ [*] | $-1.8300 \times 10^{0}$ | $2.98 \times 10^{4}$ | 3.785 |
| reemtsen1 Reemtsen (1991) [†] | $1.5249 \times 10^{-1}$ | $1.5231 \times 10^{-1}$ | $1.2689 \times 10^{5}$ | $4.974 \times 10^{2}$ |
| reemtsen2 Reemtsen (1991) [†] | $5.8359 \times 10^{-2}$ | $5.6952 \times 10^{-2}$ | $1.0145 \times 10^{5}$ | 17.249 |
| reemtsen3 Reemtsen (1991) [†] | $7.3547 \times 10^{-1}$ | $7.3548 \times 10^{-1}$ | $1.6633 \times 10^{5}$ | 15.484 |
| reemtsen4 Reemtsen (1991) [†] | $1.1401 \times 10^{-2}$ | $1.0001 \times 10^{-5}$ | $4.5109 \times 10^{5}$ | 64.185 |
| reemtsen5 Reemtsen (1991) [†] | $8.8932 \times 10^{-2}$ | $2.8761 \times 10^{-2}$ | $1.4513 \times 10^{5}$ | 1.950 |
| potchinkov3 Potchinkov (1997) [†] | - | $3.4226 \times 10^{-3}$ | - | 5.788 |
| potchinkovPL Potchinkov (1997) [†] | | $9.2940 \times 10^{-7}$ | | 2.595 |
| powell1 Todd (1994) | $-1.0000 \times 10^{0}$ | $-1.0000 \times 10^{0}$ | $9.2 \times 10^{2}$ | 3.096 |
| hettich2 Hettich (1979) [†] | $5.3800 \times 10^{-1}$ | $5.3742 \times 10^{-1}$ | $2.68 \times 10^{3}$ | 3.98 |
| hettich4 Hettich (1979) [†] | $1.0000 \times 10^{0}$ | $1.0001 \times 10^{0}$ | $3.6 \times 10^{2}$ | 8.137 |
| hettich5 Hettich (1979) [†] | $5.3800 \times 10^{-1}$ | $5.3505 \times 10^{-1}$ | $1.1995 \times 10^{5}$ | 10.909 |
| hettich6 Hettich (1979) [†] | $2.8100 \times 10^{-2}$ | $2.8163 \times 10^{-2}$ | $5.504 \times 10^{4}$ | 1.735 |
| hettich7 Hettich (1979) [†] | $1.7800 \times 10^{-1}$ | $1.7776 \times 10^{-1}$ | $4.976 \times 10^{4}$ | 7.776 |
| hettich8 Hettich (1979) [†] | - | $2.996 \times 10^{-2}$ | $2.290 \times 10^{3}$ | $1.139 \times 10^{2}$ |
| hettich9 Hettich (1979) [†] | $3.4700 \times 10^{-3}$ | $3.4791 \times 10^{-3}$ | $8.465 \times 10^{4}$ | 3.837 |
| hettich12 Hettich (1979) [†] | - | $1.0252 \times 10^{-3}$ | $7.844 \times 10^{4}$ | $8.695 \times 10^{3}$ |
| priceK Price (1992) | $-3.0000 \times 10^{0}$ | $-3.0000 \times 10^{0}$ | $5.1 \times 10^{2}$ | 3.191 |
| still1 Still (2001) | $1.0000 \times 10^{0}$ | $9.9771 \times 10^{-1}$ | $3.9 \times 10^{2}$ | 4.888 |
| userman | - | $1.2802 \times 10^{-7}$ | - | 4.028 |
| watson4a Watson (1983) | $6.4904 \times 10^{-1}$ | $6.5012 \times 10^{-1}$ | $1.78 \times 10^{3}$ | 10.116 |
| watson4b Watson (1983) | $6.1688 \times 10^{-1}$ | $6.1610 \times 10^{-1}$ | $2.22 \times 10^{3}$ | 37.654 |
| watson4c Watson (1983) | $6.1661 \times 10^{-1}$ | $6.1564 \times 10^{-1}$ | $2.82 \times 10^{3}$ | 10.029 |
| watson5 Watson (1983) | $4.3012 \times 10^{0}$ | $4.2966 \times 10^{0}$ | $8.4 \times 10^{2}$ | 17.097 |
| watson7 Watson (1983) | $1.0000 \times 10^{0}$ | $9.9777 \times 10^{-1}$ | $1.23 \times 10^{3}$ | 7.534 |
| watson8 Watson (1983) | $2.4356 \times 10^{0}$ | $2.4356 \times 10^{0}$ | $2.189 \times 10^{4}$ | $1.627 \times 10^{2}$ |
| watson10 Watson (1983) | $2.7527 \times 10^{-1}$ | $-7.0000 \times 10^{-5}$ | - | 6.952 |
| zhou1 Zhou & Tits (1996) [†] | $2.3605 \times 10^{-1}$ | $2.3505 \times 10^{-1}$ | $3.09 \times 10^{3}$ | 42.561 |

[*] Not reported in original literature but obtained via `SIPAMPL`

[†] This is a Chebyshev center problem.

M. C. Ferris and A. B. Philpott. An interior point algorithm for semi-infinite linear programming. *Mathematical Programming*, 43:257–276, 1989.

A. V. Fiacco and Yo. Ishizuka. Sensitivity and stability analysis for nonlinear programming. *Annals of Operations Research*, 27(1-4):215–235, 1990. doi: `https://doi.org/10.1007/BF02055196`.

S. Goerner. *Ein Hybridverfahren zur Loesung nichtlinearer semi-infiniter Optimierungsprobleme*. PhD thesis, Berlin University, 1997.

R. Hettich. A comparison of some numerical methods for semi-infinite programming. In R. Hettich (ed.), *Semi-infinite Programming*, volume 15 of *Lecture Notes in Control and Information Sciences*, pp. 112–125. Springer Verlag, Berlin, 1979.

W. V. Honstede. An approximation method for semi-infinite problems. In R. Hettich (ed.), *Semi-infinite Programming*, volume 15 of *Lecture Notes in Control and Information Sciences*, pp. 126–136. Springer Verlag, Berlin, 1979.

K. O. Kortanek and H. No. A central cutting plane algorithm for convex semi-infinite programming problems. *SIAM Journal on Optimization*, 3(4):901–918, 1993.

T. Leon, S. Sanmatias, and E. Vercher. On the numerical treatment of linearly constrained semi-infinite optimization problems. *European Journal of Operational Research*, 121:78–91, 2000.

C.-J. Lin, S.-C. Fang, and S.-Y. Wu. An unconstrained convex programming approach to linear semi-infinite programming. *SIAM Journal on Optimization*, 8(2):443–456, May 1998.

Zhuanghua Liu, Cheng Chen, Luo Luo, and Bryan Kian Hsiang Low. Zeroth-order methods for constrained nonconvex nonsmooth stochastic optimization. In *Forty-first International Conference on Machine Learning*, 2024.

C. A. Micchelli and T. J. Rivlin. A survey of optimal recovery. In *Optimal Estimation in Approximation Theory*, pp. 1–54. Plenum, New York, 1977.

P. Paruchuri and D. Chatterjee. Attaining the Chebyshev bound for optimal learning: a numerical algorithm. *Systems & Control Letters*, 181, 2023. paper no. 105648, doi: `https://doi.org/10.1016/j.sysconle.2023.105648`.

A. W. Potchinkov. Design of optimal linear phase fir filters by a semi-infinite programming technique. *Signal Processing*, 58:165–180, 1997.

C. J. Price. *Nonlinear Semi-infinite Programming*. PhD thesis, University of Canterbury, New Zealand, August 1992.

C. J. Price and I. D. Coope. Numerical experiments in semi-infinite programming. *Computational Optimization and Applications*, 6:169–189, 1996.

R. Reemtsen. Discretization methods for the solution of semi-infinite programming problems. *Journal of Optimization Theory and Applications*, 71(1), 1991.

G. Still. Discretization in semi-infinite programming: the rate of convergence. *Mathematical Programming*, 91:53–69, 2001.

D. M. J. Tax and R. P. W. Duin. Support vector domain description. *Pattern Recognition Letters*, 20(11-13): 1191–1199, 1999.

M. L. Todd. Interior-point algorithms for semi-infinite programming. *Mathematical Programming*, 65:217–245, 1994.

A. I. Vaz, E. M. G. P. Fernandes, and M. P. S. F. Gomes. Sipampl: Semi-infinite programming with ampl. *ACM Transactions on Mathematical Software*, 30(1):54–78, 2004. doi: 10.1145/974781.974784.

A. Ismael F. Vaz. CSIP (Convex Semi-Infinite Programming) Solver. `http://www.norg.uminho.pt/aivaz/csip.html`, 2001. University of Minho, Portugal.

Andreas Wächter and Lorenz T. Biegler. On the implementation of an interior-point filter line-search algorithm for large-scale nonlinear programming. *Mathematical Programming*, 106(1):25–57, 2006. doi: 10.1007/s10107-004-0559-y. URL `https://doi.org/10.1007/s10107-004-0559-y`.

G. A. Watson. Numerical experiments with globally convergent methods for semi-infinite programming problems. 1983. URL `https://api.semanticscholar.org/CorpusID:122452163`.

Yong Xia, Meijia Yang, and Shu Wang. Chebyshev center of the intersection of balls: complexity, relaxation and approximation. *Mathematical Programming*, 187(1):287–315, 2021.

J. L. Zhou and A. L. Tits. An SQP algorithm for finely discretized continuous minimax problems and other minimax problems with many objective functions. *SIAM Journal on Optimization*, 6(2):461–487, 1996.

K. Zhou, J. C. Doyle, and K. Glover. *Robust and Optimal Control*, volume 2. Prentice Hall, 1996.

## A   Lipschitz solutions to multiparametric programming problems

Let $v \in \mathbb{N}^*$ and let $\mathrm{L}_2(\mathbb{R}^v \times \mathbb{R}^v; \mathbb{R})$ denote the family of symmetric bilinear maps from $\mathbb{R}^v \times \mathbb{R}^v$ into $\mathbb{R}$. Recall that a mapping $\varphi : \mathbb{R}^v \to \mathbb{R}$ is *locally Lipschitz* if at each point $z \in \mathbb{R}^v$ there exists a neighborhood $\mathbb{O} \ni z$ and $L > 0$ such that $|\varphi(z') - \varphi(z'')| \leqslant L \|z' - z''\|$ whenever $z', z'' \in \mathbb{O}$. The constant $L$ depends on $z$ and $\mathbb{O}$ in general, and $L$ is the *Lipschitz modulus* of the map $\varphi$. In particular, if $\varphi$ is twice continuously differentiable, then $\varphi$ is locally Lipschitz.

Recall that a set-valued map $F : \mathbb{R}^v \rightrightarrows \mathbb{R}^{v'}$ is a mapping $F : \mathbb{R}^v \to 2^{\mathbb{R}^{v'}}$ in the standard sense; i.e., $F$ assigns to each vector $y \in \mathbb{R}^v$ a subset of $\mathbb{R}^{v'}$. Such a set-valued map is *uniformly compact* around $y \in \mathbb{R}^v$ if there is a neighborhood $O$ containing $y$ such that $\bigcup_{y' \in O} F(y')$ is bounded.

Consider the parametric nonlinear program

$$\begin{aligned} \underset{\xi}{\text{minimize}} \quad & f_\circ(\xi, \theta) \\ \text{subject to} \quad & \begin{cases} f_i(\xi, \theta) \leqslant 0 \quad \text{for } i = 1, \ldots, p, \\ \xi \in \mathbb{R}^v, \theta \in \Theta, \end{cases} \end{aligned} \qquad \text{A1}$$

along with the data

(A1.a)  $\Theta \subset \mathbb{R}^m$ is a non-empty and compact set,

(A1.b)  $f_\circ : \mathbb{R}^v \times \mathbb{R}^m \to \mathbb{R}$ is a continuously differentiable (objective) function,

(A1.c)  $f_i : \mathbb{R}^v \times \mathbb{R}^m \to \mathbb{R}$ is a continuously differentiable (constraint) function for each $i = 1, \ldots, p$.

The feasible set for A1 is the set-valued map $S_{\text{feas}} : \Theta \rightrightarrows \mathbb{R}^v$, the value of A1 is the function $\Theta \ni \theta \mapsto S_{\text{opt}}(\theta) \coloneqq \text{value of A1} \in \mathbb{R}$, and the optimizers are given by the set-valued map $S_{\text{opt}} : \Theta \rightrightarrows \mathbb{R}^v$ defined by

$$S_{\text{opt}}(\theta) = \left\{ y \in \mathbb{R}^v \,\big|\, f_\circ(y, \theta) = S_{\text{val}}(\theta) \right\}.$$

**Proposition A.1** ((Fiacco & Ishizuka, 1990, Theorem 4.2)). *Consider* A1 *along with its associated data. For $\bar{\theta} \in \Theta$ and a point $\bar{\xi} \in S_{\text{opt}}(\bar{\theta})$, let $I(\bar{\xi}) \coloneqq \left\{ i \in \{1, \ldots, p\} \,\big|\, f_i(\bar{\xi}, \bar{\theta}) = 0 \right\}$ denote the set of active constraints at $(\bar{\theta}, \bar{\xi})$. Suppose that there exists a vector $v \in \mathbb{R}^v$ such that $\left\langle \nabla_\xi f_i(\bar{\xi}, \bar{\theta}), v \right\rangle < 0$ for each $i \in I(\bar{\xi})$,[4] and that the set-valued map $S_{\text{feas}}(\cdot)$ is uniformly compact around $\bar{\theta}$. Then the function $S_{\text{val}}(\cdot)$ corresponding to* A1 *is locally Lipschitz around $\bar{\theta}$.*

## B   Convergence Analysis

In this section, we provide a brief analysis of the convergence properties of the `gradOL`. We rely on the framework of randomized smoothing and generalized Goldstein stationarity. Below, we cite the relevant definitions and auxiliary results from Liu et al. (2024).

### B.1   Definitions and Notations

**Notation.** Let $\mathcal{X} \subset \mathbb{R}^{n+1}$ be a convex and compact set with diameter bounded by $B$. As before, we denote the Euclidean norm by $\|\cdot\|$.

Note that the outer-level objective is to maximize $\mathcal{G}(\bar{u})$ (see (9)) with respect to the uncertainty variable $\bar{u}$. To retain generality while simplifying notation, we introduce a slight abuse of notation and define $F \coloneqq -\mathcal{G}$. We also relabel the uncertainty variable $\bar{u}$ as $x$ preserving the structure of the original problem while only modifying symbols. Since `gradOL` seeks to maximize $\mathcal{G}(\bar{u})$ over $\bar{u}$, this transformation yields the equivalent minimization problem

$$\min_{x \in \mathcal{X}} F(x).$$

---

[4]In other words, the Mangasarian-Fromovitz constraint qualification conditions hold at $\bar{\xi}$.

**Smoothed Objective.** Consider the problem $\min_{x \in \mathfrak{X}} F(x)$, where $F$ is locally Lipschitz as established in Theorem 3.1 but potentially nonconvex and nonsmooth. We utilize the $\delta$-smoothed approximation:

$$F_\delta(x) := \mathbb{E}_{u \sim \text{Unif}(B(0,1))}\big[F(x + \delta u)\big].$$

As established in Liu et al. (2024), $F_\delta$ is differentiable with a Lipschitz continuous gradient. Specifically, if $F$ is $G$-Lipschitz, then $F_\delta$ is $L_\delta$-smooth with constant $L_\delta = \frac{cG\sqrt{n+1}}{\delta}$ for some dimension-dependent constant $c$.

**Generalized Gradient Mapping.** For a parameter $\gamma > 0$, a point $x \in \mathfrak{X}$, and a gradient vector $v \in \mathbb{R}^{n+1}$, the generalized gradient mapping is defined as:

$$\mathbb{G}(x, v, \gamma) := \frac{1}{\gamma}\left(x - \arg\min_{y \in \mathfrak{X}} \left\{\langle v, y \rangle + \frac{1}{2\gamma}\|y - x\|^2\right\}\right).$$

This mapping serves as a proxy for stationarity in constrained optimization.

## B.2 Preliminaries and Assumptions

**Lemma B.1** (Gradient Bound). *If $h : \mathbb{R}^{n+1} \to \mathbb{R}$ is $L$-Lipschitz, then $\|\nabla h(x)\| \leq L$ wherever the gradient exists.*

**Assumption 1** (Inexact Gradient). At any query point $x_R$, the algorithm receives an inexact gradient estimator $v_R$ satisfying:

$$\|v_R - \nabla F_\delta(x_R)\| \leq e.$$

**Assumption 2** (Lipschitz Continuity). The function $F_\delta$ is $G$-Lipschitz on $\mathfrak{X}$. Consequently, by standard properties (Lemma B.1), $\|\nabla F_\delta(x)\| \leq G$ for all $x \in \mathfrak{X}$.

## B.3 Main Convergence Result

We analyze the output $F(x_R)$, where $R$ is drawn uniformly from $\{0, \ldots, T-1\}$, of `gradOL` and $T$ is the number of iterations.

**Theorem B.1.** *Let step size $\gamma = \frac{\delta}{cG\sqrt{n+1}}$. Under the stated assumptions, if the number of iterations $T$ satisfies*

$$T \geq \frac{4G^2 Bc\sqrt{n+1}}{\delta(\epsilon^2 - 4G^2 - 8e^2)},$$

*assuming the denominator is positive, then the iterates satisfy the expected stationarity bound $\mathbb{E}[\|\mathbb{G}(x_R, \nabla F_\delta(x_R), \gamma)\|] \leq \epsilon$.*

*Proof.* **Step 1: Descent Lemma with Inexact Gradients.** Since $F_\delta$ is $L_\delta$-smooth with $L_\delta = \frac{cG\sqrt{n+1}}{\delta}$, the standard descent lemma implies:

$$F_\delta(x_{t+1}) \leq F_\delta(x_t) + \langle \nabla F_\delta(x_t), x_{t+1} - x_t \rangle + \frac{L_\delta}{2}\|x_{t+1} - x_t\|^2$$

$$= F_\delta(x_t) - \gamma\langle \nabla F_\delta(x_t), \mathbb{G}(x_t, v_t, \gamma)\rangle + \frac{L_\delta \gamma^2}{2}\|\mathbb{G}(x_t, v_t, \gamma)\|^2.$$

Substituting $v_t$ into the inner product:

$$-\langle \nabla F_\delta(x_t), \mathbb{G}(x_t, v_t, \gamma)\rangle = -\langle v_t, \mathbb{G}(x_t, v_t, \gamma)\rangle + \langle v_t - \nabla F_\delta(x_t), \mathbb{G}(x_t, v_t, \gamma)\rangle.$$

Using the property of the gradient mapping that $-\langle v_t, \mathbb{G}(x_t, v_t, \gamma)\rangle \leq -\|\mathbb{G}(x_t, v_t, \gamma)\|^2$ (Lemma C.2 Liu et al. (2024)), we have:

$$F_\delta(x_{t+1}) \leq F_\delta(x_t) - \left(\gamma - \frac{L_\delta \gamma^2}{2}\right)\|\mathbb{G}(x_t, v_t, \gamma)\|^2 + \gamma\langle v_t - \nabla F_\delta(x_t), \mathbb{G}(x_t, v_t, \gamma)\rangle.$$

**Step 2: Error Decomposition.** We bound the inner product term using Cauchy-Schwarz and Young's inequality :

$$\langle v_t - \nabla F_\delta(x_t), \mathbb{G}(x_t, v_t, \gamma)\rangle = \langle v_t - \nabla F_\delta(x_t), \mathbb{G}(x_t, \nabla F_\delta(x_t), \gamma)\rangle$$
$$+ \|v_t - \nabla F_\delta(x_t)\|\|\mathbb{G}(x_t, v_t, \gamma) - \mathbb{G}(x_t, \nabla F_\delta(x_t), \gamma)\|$$
$$\leqslant \langle v_t - \nabla F_\delta(x_t), \mathbb{G}(x_t, \nabla F_\delta(x_t), \gamma)\rangle + \|v_t - \nabla F_\delta(x_t)\|^2$$

Using Cauchy–Schwarz and Young's inequality on the RHS above

$$\langle v_t - \nabla F_\delta(x_t), \mathbb{G}(x_t, v_t, \gamma)\rangle \leqslant \frac{3}{2}\|v_t - \nabla F_\delta(x_t)\|^2 + \frac{1}{2}\|\mathbb{G}(x_t, \nabla F_\delta(x_t), \gamma)\|^2.$$

Taking expectations and applying Assumption 1:

$$\left(\gamma - \frac{L_\delta \gamma^2}{2}\right)\mathbb{E}[\|\mathbb{G}(x_t, v_t, \gamma)\|^2] \leqslant \mathbb{E}[F_\delta(x_t) - F_\delta(x_{t+1})] + \frac{3\gamma}{2}e^2 + \frac{\gamma}{2}\mathbb{E}[\|\mathbb{G}(x_t, \nabla F_\delta(x_t), \gamma)\|^2].$$

**Step 3: Telescoping Sum.** Summing from $t = 0$ to $T - 1$, dividing by $T$, and using $\gamma = 1/L_\delta$ (specifically $\gamma = \frac{\delta}{cG\sqrt{n+1}}$) such that the coefficient on the LHS becomes $\gamma/2$:

$$\frac{\gamma}{2}\mathbb{E}[\|\mathbb{G}(x_R, v_R, \gamma)\|^2] \leqslant \frac{F_\delta(x_0) - F_\delta(x_T)}{T} + \frac{3\gamma}{2}e^2 + \frac{\gamma}{2}\mathbb{E}[\|\mathbb{G}(x_R, \nabla F_\delta(x_R), \gamma)\|^2].$$

Rearranging and bounding $F_\delta(x_0) - F_\delta(x_T) \leqslant GB$:

$$\mathbb{E}[\|\mathbb{G}(x_R, v_R, \gamma)\|^2] \leqslant \frac{2GB}{T\gamma} + 3e^2 + \mathbb{E}[\|\mathbb{G}(x_R, \nabla F_\delta(x_R), \gamma)\|^2]. \tag{A2}$$

**Step 4: Final Bound.** We bound the true gradient mapping term on the RHS using the Lipschitz constant $G$ (using $\|\mathbb{G}(x, \nabla F, \gamma)\| \leqslant \|\nabla F\| \leqslant G$ using Lemma B.1). Substituting this into A2:

$$\mathbb{E}[\|\mathbb{G}(x_R, v_R, \gamma)\|^2] \leqslant \frac{2GB}{T\gamma} + 3e^2 + G^2.$$

Finally, we relate the true mapping back to the approximate mapping:

$$\mathbb{E}[\|\mathbb{G}(x_R, \nabla F_\delta(x_R), \gamma)\|^2] \leqslant 2\mathbb{E}[\|v_R - \nabla F_\delta(x_R)\|^2] + 2\mathbb{E}[\|\mathbb{G}(x_R, v_R, \gamma)\|^2]$$
$$\leqslant 2e^2 + 2\left(\frac{2GB}{T\gamma} + 3e^2 + G^2\right)$$
$$= \frac{4GB}{T\gamma} + 4G^2 + 8e^2.$$

We obtain the sufficient condition for the squared norm to be bounded by $\epsilon^2$:

$$\frac{4GB}{T\gamma} + 4G^2 + 8e^2 \leqslant \epsilon^2 \implies T \geqslant \frac{4GB}{\gamma(\epsilon^2 - 4G^2 - 8e^2)}.$$

Substituting $\gamma = \frac{\delta}{cG\sqrt{n+1}}$ yields the stated result. $\qquad\square$

## C  Iterative Sampling Routine

By sampling $N$ many $u(s)$ from $K$ we replace the infinite constraint in (11) by maximization over $N$ constrained minimization problems, solve the resulting convex program to get $s^j$, and record $f_N^j$. Repeating $M$ times yields an empirical distribution of $f_N$; we accept $N$ once the fraction of trials within $\delta$ of the true optimum $f^*$ reaches 80%.

---

**Algorithm 2** Iterative Sampling for CSIP (11)

---

**Require:** $N, M \in \mathbb{N}$, tolerance $\delta > 0$, true value $f^*$

1: **for** $j = 1, \ldots, M$ **do**
2:    Draw i.i.d. samples $\{u^i\}_{i=1}^N \subset K$
3:    Solve

$$f_N^j = \min_{(s,x) \in \mathbb{R} \times X} s \quad \text{s.t.} \quad \|u^i - x\|^2 - s \leq 0 \; \forall i = 1, \ldots, N$$

4: **end for**
5: Compute empirical confidence

$$\hat{p}_N = \frac{1}{M} \sum_{j=1}^M \mathbf{1}(|f_N^j - f^*| \leq \delta).$$

6: **if** $\hat{p}_N \geq 0.8$ **then**
7:    $N$ is sufficient
8: **else**
9:    Increase $N$ and repeat
10: **end if**

---

## D  Hyperparameters

Hyperparameters for `gradOL` are chosen after experimenting with the benchmark problems. We observe that runtimes are somewhat robust to some of the given parameters, which can be chosen as per the problem at hand.

Table 3: Hyperparameters for `gradOL`

| Category | Value |
|---|---|
| Learning rate ($\eta$) | $1 \times 10^{-1}$ |
| Tolerance ($\delta$) | $1 \times 10^{-3}$ |
| Max Epochs (`M`) | $1 \times 10^4$ |
| Barrier Parameter ($\alpha$) | $1 \times 10^5$ |

For MSA-Simulated Annealing, we use Ipopt Optimizer Wächter & Biegler (2006) for the inner minimization, the hyperparameters are given in 4. As described in the routine we adapt for Iterative Sampling in C the hyperparameters are given in 5.

Table 4: MSA–Simulated Annealing

| Category | Value |
|---|---|
| Maximum iterations | $1 \times 10^4$ |
| Initial temperature | 1.0 |
| Cooling rate | $9.95 \times 10^{-1}$ |

Table 5: Iterative Sampling

| Category | Value |
|---|---|
| Maximum iterations ($M$) | $1 \times 10^2$ |
| Number of Random Samples ($N$) | $10 - 1 \times 10^4$ |
| Confidence Threshold | $8 \times 10^{-1}$ |
| Tolerance ($\delta$) | $1 \times 10^{-3}$ |

## E  Ablation Study

We report separate ablation tables for five representative Chebyshev center problems, examining variations in learning rates and barrier parameters. The comparisons in Table 6 highlight the sensitivity of `gradOL` to these hyperparameters. Notably, `gradOL` relies on only two scalar hyperparameters, each offering an intuitive sense of how the algorithm's behavior changes as the parameters are adjusted. In particular, reducing the learning rate from its default setting ($\eta = 10^{-1}$)typically leads to slower convergence. Conversely, while

larger values of $\eta$ may accelerate convergence, they often induce instability in the form of oscillatory or *zig-zag* optimization trajectories.

On the other hand, the choice of the barrier parameter $\alpha$ directly affects the accuracy of the gradient estimates required for the outer maximization of $\mathcal{G}$. In particular, $\alpha$ must be sufficiently large to ensure that the inner minimization is solved with high fidelity, thereby yielding reliable gradient information for the outer problem. This effect is also reflected in our ablation study.

Our ablation study considers five benchmark representation problems: `leon9`, `kortanek1`, `leon10`, `reemtsen3`, and `hettich2`. Unless otherwise specified, we use the default parameter setting $(\eta, \alpha) = (0.1, 10^5)$. We then conduct a simple grid search over learning rates $0.01, 0.1, 0.5$ and penalty parameters $10^4, 10^5, 10^6$.

As shown in Table 6, the results corroborate the expectation that larger values of $\alpha$ yield more accurate gradient estimates, while smaller learning rates improve convergence at the cost of increased runtime. The problems exhibit stronger sensitivity to $\alpha$, indicating that the ability to compute accurate gradients (controlled by $\alpha$) largely determines whether the optimization succeeds. In contrast, the adverse effects of reducing the learning rate can, in principle, be offset by longer runtimes. However, when $\alpha$ is insufficiently small, the optimizer may fail to identify the correct solution and can miss the optimum entirely.

Table 6: (Obtained Value / Runtime) for different Chebyshev Center problems

**leon9**

| $\alpha$ \ $\eta$ | $10^{-2}$ | $10^{-1}$ | $5 \times 10^{-1}$ |
|---|---|---|---|
| $10^4$ | 0.1623
$4.259 \times 10^2$ ms | 0.1644
56.573 ms | $1 \times 10^{-6}$
$8.32 \times 10^2$ ms |
| $10^5$ | 0.1421
55.837 ms | 0.16338
8.694ms | 0.1421
$8.466 \times 10^2$ ms |
| $10^6$ | 0.0856
842.124 ms | 0.0849
55.451 ms | 0.1128
54.495 ms |

Optimal Value = 0.16338

**kortanek1**

| $\alpha$ \ $\eta$ | $10^{-2}$ | $10^{-1}$ | $5 \times 10^{-1}$ |
|---|---|---|---|
| $10^4$ | 3.2175
2.028 ms | 3.1198
2.528 ms | 3.1198
2.7552 ms |
| $10^5$ | 2.9032
1.782 ms | 3.2184
6.018 ms | 3.0004
7.011 ms |
| $10^6$ | 2.8152
1.784 ms | 3.2211
1.819 ms | 3.2052
1.802 ms |

Optimal Value = 3.2212

**leon10**

| $\alpha$ \ $\eta$ | $10^{-2}$ | $10^{-1}$ | $5 \times 10^{-1}$ |
|---|---|---|---|
| $10^4$ | $1 \times 10^{-6}$
6.926 ms | $1 \times 10^{-6}$
6.051 ms | $1 \times 10^{-6}$
6.104 ms |
| $10^5$ | 0.4778
20.704 ms | 0.5377
19.612 ms | $1 \times 10^{-6}$
6.166 ms |
| $10^6$ | 0.5372
143.222 ms | $1 \times 10^{-6}$
11.366 ms | $1 \times 10^{-6}$
6.439 ms |

Optimal Value = 0.5382

**reemtsen3**

| $\alpha$ \ $\eta$ | $10^{-2}$ | $10^{-1}$ | $5 \times 10^{-1}$ |
|---|---|---|---|
| $10^4$ | 0.7125
$4.46 \times 10^4$ ms | $1 \times 10^{-6}$
$3.87 \times 10^2$ ms | $1 \times 10^{-6}$
40.726 ms |
| $10^5$ | 0.7352
$1.16 \times 10^3$ ms | 0.7354
15.484 ms | $1 \times 10^{-6}$
$3.89 \times 10^2$ ms |
| $10^6$ | 0.7351
$1.34 \times 10^4$ ms | 0.7352
$1.307 \times 10^3$ ms | $1 \times 10^{-6}$
$52.71 \times 10^3$ ms |

Optimal Value = 0.7354

**hettich2**

| $\alpha$ \ $\eta$ | $10^{-2}$ | $10^{-1}$ | $5 \times 10^{-1}$ |
|---|---|---|---|
| $10^4$ | 0.48513
2.544 ms | 0.47269
2.017 ms | 0.0001
2.174 ms |
| $10^5$ | 0.5355
7.826 ms | 0.53742
3.98 ms | 0.5372
67.937 ms |
| $10^6$ | 0.5307
$3.702 \times 10^4$ ms | 0.5378
$1.4 \times 10^4$ ms | 0.53834
$8.579 \times 10^3$ ms |

Optimal Value = 0.538

