# OpenReview forum: "On a Gradient Approach to Chebyshev Center Problems with Applications to Function Learning"
_TMLR — Accepted by TMLR_

### Review · Reviewer_bE3N · 2025-10-31

**Summary Of Contributions:**

The paper proposes a gradient-based algorithm to solve the Chebyshev center problem via a novel equivalent finite-dimensional formulation of the original problem and the auto-differentiation algorithm. The local Lipschitz continuity of the objective function is also proved to justify the use of the (sub)gradient-based optimization algorithm. In contrast, most of the existing works in the literature solve some relaxed version of the original problem.

**Additional Comments:**

NA

**Audience:**

Yes

**Audience Explanation:**

People who are working on the gradient-based optimization and function learning may find this paper interesting

**Claims And Evidence:**

Yes

**Claims Explanation:**

The paper states the claims with theorems with proof, detailed algorithmic framework, and clear presentation.

**Requested Changes:**

1. Why the authors include ``optimal function learning" in the title since the main problem considered in this paper is just the Chebyshev Center problem.

2. The key advancement lies in the new formulation of the problem, i.e., Eq. (5), which is a consequence of the recalled Theorem 2.1 in the paper. This should be mentioned early in the introduction and in the abstract.

3. The sections are not separated properly. For example, the reformulation of the problem (Eq. (9)) could be put in Section 2. In addition, the section titles are too abstract and should be made more explicit.

4. The Lipschitz continuity is valid when the inner optimisation problem is solved exactly. Will the implemented optimization solver in Algorithm 1 influence this property?

---

> ### Author Response · Authors · 2025-11-13
>
> We sincerely thank the reviewer for their careful reading of our manuscript and for providing insightful and constructive feedback.
> ## 1. On the use of "optimal function learning" in the title:
> We thank the reviewer for this important question. The title was chosen because the Chebyshev center problem provides the precise mathematical formulation for optimal learning from data, particularly in a non-noisy setting. As established in the foundational work of Micchelli & Rivlin (1977) and reinforced in the context of modern learning theory by Binev *et al.* (2024), the task of optimal learning can be framed in principle as follows:
>
> Given a set of observations, we first identify the *hypothesis space* $K$ of all functions that are consistent with this data. The goal is to select a single function $f$ (*optimal interpolant*) from a model class $X$ that minimizes the worst-case error with respect to any other valid hypothesis $g$ in $K$. This is exactly the variational problem defining the Chebyshev center:
>
> $$
> \min_{f \in X} \, \sup_{g \in K} \, \| f - g \|.
> $$
>
> Our algorithm operates in the finite-dimensional space $\mathbb{R}^n$ so that the problem stays computationally viable; see the discussion in Alimov & Tsar'kov's book *Geometric Approximation Theory*, 2021, Section 16.1. The vectors in $\mathbb{R}^n$ represent the coefficients or discretized values of functions within a finite-dimensional function space. Therefore, by solving the Chebyshev center problem in $\mathbb{R}^n$, our algorithm directly computes the solution to the optimal function learning problem in this practical finite-dimensional setting.
>
> ##  2. On highlighting the key advancement earlier:
> The reviewer is correct that the reformulation of the semi-infinite problem into an entirely finitary, albeit nested, optimization problem (Eq. 5) is the central technical lever that enables our gradient-based approach. We agree that this point should be emphasized earlier to better frame our contribution. In our revision, we will make the following changes:
> 1. In the Abstract: We will add a sentence to explicitly state this reformulation. For instance:
> "Our approach is enabled by a key theoretical result that recasts the semi-infinite problem into a finite-dimensional max-min structure, making it amenable to modern automatic differentiation frameworks for the first time."
> 2. In the Introduction: We will introduce this idea in the first few paragraphs.
>
> ## 3. On section structure and titles:
> We thank the reviewer for this valuable feedback on the paper's organization and readability. In the revised version, we will implement the following restructuring:
> 1. We will follow your suggestion and move the specific application of Theorem 2.1 to the Chebyshev center problem, which yields the crucial formulation in Eq. (9), to the end of Section 2 (Preliminaries). This will logically group the foundational theory with its direct application to our problem of interest.
> 2. We will rename Section 3 from the general title "The Key Advancement" to something more descriptive, such as "A Differentiable Max-Min Formulation and the gradOL Algorithm". This new section will then focus squarely on the properties of the reformulated problem (i.e., the Lipschitz continuity proof) and the detailed presentation of our proposed algorithm, gradOL.
>
> ## 4. On the influence of inexact inner-loop optimization:
> We thank the reviewer for raising this important point. You are correct that our formal proof of the Lipschitz continuity of the objective function $\mathcal{G}$ (Theorem 3.1) assumes that the inner minimization problem is solved to exact optimality. In our practical implementation, we use iterative, off-the-shelf convex solvers which find an approximate solution up to a specified numerical tolerance. This discrepancy introduces an error in the evaluation of $\mathcal{G}$, and consequently, the gradient computed via automatic differentiation is an *approximation* of the true sub-gradient. The influence of this approximation is twofold:
>
> 1. While the rigorous proof of Lipschitz continuity doesn't directly apply to the numerically computed function, the underlying stability of the problem, guaranteed by properties like the MFCQ (as mentioned in Step 3 of the proof of Theorem 3.1), ensures that small perturbations in the input $\bar{u}$ lead to small changes in the optimal value. Therefore, as the accuracy of the inner solver increases, the computed objective function $\mathcal{G}(\bar{u})$ approaches the true function, and the inexact gradient approaches the true gradient.
>
> 2.  In our implementation, we mitigate the impact of this approximation by setting a very high accuracy (a small tolerance $\delta$) for the inner convex solver. Furthermore, our preliminary complexity analysis explicitly incorporates a term $\varepsilon$ for the error in the gradient approximation, showing that convergence can still be characterized even in the presence of such numerical inaccuracies.

---

> > ### Author Response · Authors · 2025-11-21
> > **Revision Update**
> >
> > We thank you for your valuable feedback. The changes you requested have been incorporated into the revised manuscript. We invite you to review the updated version and the rebuttal. We are available to address any follow-up questions.

---

### Review · Reviewer_LH3U · 2025-11-05

**Summary Of Contributions:**

This paper leverages modern automatic differentiation techniques to exploit gradient-based optimization to directly solve the Chebyshev center problem arising in optimal function learning and convex semi-infinite programming (CSIP). The main contributions claimed include: 1) Firstly introduce the gradient-based framework gradOL for Chebyshev center problems while previous methods being mostly sampling, relaxation or non-gradient.
2) Theoretically proves optimality the Chebyshev center problem of under strong convexity.
3) Empirically demonstrate the accuracy and efficiency of gradOL on broad benchmarks.

Key strengths: novel idea of leveraging autodiff for CSIP/Chebyshev center; broad empirical results; practical implementation; relevance to optimization and learning community

Key weaknesses: assumptions (e.g., strong convexity) may limit generality; lack of ablation studies: The sensitivity to hyperparameters (learning rate $\eta$, barrier parameter $\alpha$) is discussed qualitatively but not systematically analyzed.

**Audience:**

Yes

**Audience Explanation:**

The Chebyshev central problem addressed in this paper has numerous applications, including data fitting and robust optimization (as the paper states). Furthermore, this paper introduces a novel and practical gradient-based solver, which is likely to pique the interest of researchers in machine learning, optimization, and computational mathematics within the TMLR community.

**Broader Impact Concerns:**

No direct ethical risks are apparent.

**Claims And Evidence:**

Yes

**Claims Explanation:**

1) The paper reports extensive benchmarking indicating significant speedups and good performance.

2) The paper provides proofs of Lipschitz continuity and convergence under strong convexity (as claimed) in Theorem 3.1.

**Requested Changes:**

1) Add ablation studies on the influence of hyperparameters.

2) Clarify the precise assumptions under which your method is guaranteed to work (e.g., strong convexity, smoothness, differentiability of supremum-operator). Discuss scenarios where these assumptions may not hold and how method behaves. Ideally, discuss these in a separate section.

3) It needs to be clarified in the abstract or introduction that what is being solved is not the original Chebyshev center problem, but an approximation of its finite-dimensional version.

4) Table 1 can be placed at the end of the article or in the appendix.

5) Add preliminary about automatic differentiation in the Preliminaries.

All suggestions above will simply strengthen the work in my personal view.

---

> ### Author Response · Authors · 2025-11-13
>
> We thank the reviewer for their insightful feedback.
>
> ## **1. On Ablation Studies for Hyperparameters**
>
> We agree that an ablation study is an important addition. We will add a new section to the Appendix analyzing the sensitivity of `gradOL` to its key hyperparameters: the **learning rate ($\eta$)** and the **barrier parameter ($\alpha$)**. This study will be performed on a representative subset of the benchmark problems and will provide practical guidance on hyperparameter tuning. Notably, `gradOL` relies on only two scalar hyperparameters, each offering an intuitive sense of how the algorithm’s behavior changes as the parameters are adjusted.
>
> ## **2. On Assumptions and Limitations**
>
> The assumptions in our work are natural and fairly general.
>
> - **Strong convexity** of the norm is a standard assumption unless specific properties such as sparsity, etc., are demanded; we respectfully disagree that strong convexity of the norm assumed in our work is a significant loss of generality. Moreover, the results of https://doi.org/10.1016/j.sysconle.2023.105648 transparently demonstrates a procedure to tackle the not-strongly-convex case of the norm in the Chebyshev center problem, which in turn relies on a strongly convex regularizer. In other words, treating the strongly convex case suffices.
> - **All hypotheses** needed for the technical results to proceed have been included in the manuscript.
> - **Sensitivity to hyperparameters** will be included in the revised version in the **Appendix**.
>
>
> ## **3. On Clarifying the Problem Formulation**
>
> We thank the reviewer for giving us the opportunity to clarify the problem formulation. No finitary technique is capable of solving the general infinite-dimensional Chebyshev center problem; with finite memory and finite precision arithmetic, one must employ approximations. We point the reviewer to Alimov and Tsarkov's book _Geometric Approximation Theory_, Section 16.1, for a discussion. The fact that a finite-dimensional version of the Chebyshev center problem is our target will be clarified in the Introduction. This finite-dimensional version, however, is in principle solved exactly by `gradOL` upto the precision of the computing machine, and this will also be stated in the same vein and location. All assumptions needed have been listed precisely at the necessary places.
>
>
> ## **4. On the Placement of Table 1**
>
> We agree with the reviewer. To improve the flow and readability of the main text, we will move the large benchmark table (Table 1) to the end of the main text.
>
>
> ## **5. On Adding a Preliminary on Automatic Differentiation**
>
> This is a helpful suggestion. We will add a new subsection to the **Preliminaries** section that briefly explains the principles of **automatic differentiation (AD)**. This will clarify how our algorithm computes exact gradients for the programmatically defined value function $\mathcal{G}$ without requiring an analytical formula, making the paper more self-contained.
>
>
>
> We thank the reviewer again for their constructive comments and are confident that these revisions will substantially improve our paper.

---

> > ### Author Response · Authors · 2025-11-21
> > **Revision Update**
> >
> > We thank you for your valuable feedback. The changes you requested have been incorporated into the revised manuscript. We invite you to review the updated version and the rebuttal. We are available to address any follow-up questions.

---

### Review · Reviewer_L2BP · 2025-11-06

**Summary Of Contributions:**

This manuscript clearly defines and motivates the research problem, the Chebyshev center problem by situating it within the context of prior work. The theoretical analysis is aptly initiated from the foundation of the finitary problem, as defined by Paruchuri & Chatterjee (2023). A key theoretical contribution of the paper is the detailed proof of the Lipschitz continuity of the inner objective function with respect to the outer variable. This proof is crucial as it forms the theoretical foundation for the proposed gradOL algorithm. Building upon this theoretical groundwork, the work effectively employs automatic differentiation tools for gradient computation and provides a complete description of the algorithm. The study then conducts extensive benchmarking on a series of Chebyshev center problems, convincingly demonstrating the algorithm's accuracy and marked efficiency. Furthermore, the applicability of the method is successfully extended to the more general class of Convex Semi-Infinite Programming (CSIP) problems, with experimental results robustly validating its high performance in this broader context.

This manuscript presents a highly competent and valuable contribution to the field of optimization. The problem addressed in this paper is classical, while the proposed solution demonstrates a notable degree of innovation. The theoretical foundation is laid out with commendable rigor, particularly in establishing the Lipschitz continuity.

**Audience:**

Yes

**Audience Explanation:**

The paper introduce a novel approach to solve Chebyshev center problem. The Chebyshev center problem is a significant and long-standing problem in many fields.First, the Chebyshev center problem is valuable in functional learning. And beyond the field, the manuscript itself also mentions the application of the Chebyshev center problem can involve:
robust optimization for managing uncertainty, sensor network localization for estimating node positions and location problems to optimize placement strategies.

This reviewer believe that the research can attract the interests of some people.

**Broader Impact Concerns:**

No concerns

**Claims And Evidence:**

Yes

**Claims Explanation:**

The paper provides a detailed and accurate proof of the Lipschitz continuity, which is both the core theoretical conclusion of the paper and the crucial foundation for the core algorithm of the research. The proof is presented with such thoroughness and precision that its validity is clear. This high standard of rigor enhances the confidence in the paper's theoretical findings.

The algorithmic evaluation is rigorous, as it employs a substantial set of justified metrics across a sufficient number of comparable problems and includes comparisons with a wide array of existing methods. This is supported by detailed tables of experimental results and specific discussions thereof.

Overall, in my opinion,  the claims made in the submission supported by accurate, convincing and clear evidence.

**Requested Changes:**

There are some issues that need to be addressed before it can be considered for publication. If the following problems are well-addressed, this reviewer believes that the essential contribution of this paper are important for the optimization, control and so on.

There is at least one Spelling error in the manuscript, such as, in page 6, under the Theorem 2.1, “\\( ]0,+\infty[ \\)” would be “\\( [0,+\infty] \\)”. And “functionsn” would be “functions”. Please check the manuscript carefully. This manuscript requires careful editing and particular attention to English spelling.

And in page 7, Theorem 3.1, “the value of (10)” may not be equal to “\\( (\\sup_{\\bar{u} \\in K^{n+1}} \\mathscr{G}(\\bar{u}) )^{\\frac{1}{2}} \\)”.  “the value of (10)” is the square of the Chebyshev radius but that value in Theorem 3.1 seems to be the Chebyshev radius itself. And also please check other parts of the manuscript to find whether there are similar issues.

Regarding the computational complexity of gradOL, I appreciate that a full convergence analysis is beyond the scope of this work. However, since the paper have provided a preliminary complexity bound, it would be valuable if you could include a brief sketch of the analysis or the key ideas adapted from Liu et al. (2024) to support the plausibility of this claim.

---

> ### Author Response · Authors · 2025-11-13
>
> We sincerely thank the reviewer for their constructive feedback, which has helped us improve the quality of our manuscript. We have addressed all the points raised as follows.
>
> ## 1. On Spelling and Typographical Errors:
> We are grateful to the reviewer for their careful reading. We have corrected the specific typos mentioned (`[0, +∞[` and `functionsn`) and have performed a thorough proofreading of the entire manuscript to address any remaining errors.
>
> ## 2. On the Mathematical Statement in Theorem 3.1:
>
> Thank you for the correction: the value of problem (10) corresponds to the *square* of the Chebyshev radius, which is precisely $\sup \mathcal{G}(\bar{u})$. The statement in Theorem 3.1 will be corrected in the revised manuscript, and we will verify the consistency of this correction throughout the paper.
>
> ## 3. On the Computational Complexity Sketch:
> This is an excellent suggestion. While a full convergence proof is beyond the scope of this work, we agree that a sketch of the analysis adds significant value. In the revised manuscript, we will add a brief discussion on this topic in the Appendix as requested. This new segment outlines how the complexity bound in Remark 3.2 is derived by adapting the theoretical framework from Liu et al. (2024) to our bilevel optimization structure, leveraging the Lipschitz continuity of $\mathcal{G}$ established in Theorem 3.1.

---

> > ### Author Response · Authors · 2025-11-21
> > **Revision Update**
> >
> > We thank you for your valuable feedback. The changes you requested have been incorporated into the revised manuscript. We invite you to review the updated version and the rebuttal. We are available to address any follow-up questions.

---

### Author Response · Authors · 2025-11-21
**Summary of Revisions**

We thank the reviewers for their valuable feedback. We have responded individually to all comments, and all suggested changes have been incorporated into the revised manuscript. The updated version includes an ablation study, a sketch of the convergence proof in the appendix, corrections to typographical errors, and improvements to the logical flow. All revisions are highlighted in blue. We remain available for any further questions or clarifications.

---

### Decision · Action_Editor_VbFK · 2025-12-20

**Recommendation:** Accept with minor revision

**Additional Comments:**

The reviewers provided sufficiently detailed feedback to the paper, and after the rebuttal, all three reviewers recommended acceptance, which I agree with.

Additional comments and suggestions from my side:

- One of the reviewers suggested a title change to downweight "function learning". While the authors provided reasonable justifications in the rebuttal, personally I am leaning towards the reviewer's side, especially since the technical contributions of the paper are somewhat independent of any specific function learning application, and the experiments in the paper are also not of that flavor.

- I find Remark 3.2 confusing, particularly regarding where the nonconvexity comes from. It is noted that the convergence analysis is challenging since the objective function of the inner optimization subroutine can be nonconvex and nonsmooth, therefore only the convergence to stationary points rather than to the global minimum is sketched. However, the original problem prior to the minimax reformulation is convex, suggesting that convergence to the global minimum should be intuitively attainable unless I missed some key intricacy here (in particular, around Theorem 2.1). So the questions would be, where does the nonconvexity come from, and is it necessary? This is not obvious for readers such as myself, who are unfamiliar with the line of prior works this paper builds on. Furthermore, I would suggest discussing the nonconvexity issue more thoroughly in earlier sections of the paper.

**Audience:**

Yes

**Audience Explanation:**

The Chebyshev center problem is a classical protocol in optimization and machine learning. Making contributions to it, either theoretical or empirical, should be of interest to the audience of TMLR.

**Claims And Evidence:**

Yes

**Claims Explanation:**

The submitted paper claims to make two primary contributions to solving Chebyshev center problems. First, building on an earlier result of Paruchuri and Chatterjee, the paper provides an equivalent minimax formulation of the Chebyshev center problem, and then employs efficient first-order methods to solve it. Second, while an entirely end-to-end performance guarantee appears to be out of reach, the paper theoretically justifies the use of first-order methods by proving a Lipschitz condition of the objective function.

The claimed contributions are sufficiently supported by evidence. Specifically, the proposed algorithm is evaluated in comprehensive experiments, and in response to the reviewers' feedback, a good amount of ablation study is added to the appendix. The obtained empirical results are in favor of the proposed algorithm. In addition, as evaluated by the reviewers, the proof of the Lipschitz condition is solid and technically nontrivial.

---

> ### Author Response · Authors · 2025-12-29
>
> Dear Action Editor,
>
> We thank you for accepting the paper and for the additional helpful suggestions.
>
> **(1) Title.** We appreciate the reviewer's suggestion to downweight “function learning.” While our technical contributions are broadly applicable beyond that specific setting, we note that many problems in our benchmark are indeed function approximation/learning instances (e.g., **leon1**, **goerner4**, **hettich2**, etc.). With that in mind, we will revise the title to **"On a Gradient Approach to Chebyshev Center Problems with Applications to Function Learning”**.
>
> **(2) Clarification on Remark 3.2 (source of nonconvexity).** We believe there may be a misunderstanding here. The original formulation is a convex optimization problem (with a unique minimizer under the stated assumptions), but it involves **infinitely many constraints**.
>
> To enable a fully finite-dimensional, gradient-based procedure, we use the min–max reformulation in Theorem 2.1, which preserves the optimal value of the original problem. The key point is:
>
> For any fixed outer variable $u$, the **inner minimization remains convex** (it is an infimum of a convex objective over a convex feasible region).
> The **outer maximization over $u$**, however, is generally **non-concave and potentially non-smooth**, which is precisely what Remark 3.2 addresses. This is why the analysis naturally discusses convergence to (generalized) stationary points rather than global maximizers of the reformulated objective.
>
> Finally, Theorem 2.1 concerns the **optimal value equivalence** between the original convex problem and the reformulated max–min problem. While the primal minimizer of the original problem is unique, the outer maximization can admit **many maximizers**, in particular, there can be up to **\(n!\)** such maximizers arising from symmetry/permutations of active tuples.

---

> > ### Author Response · Authors · 2026-01-05
> > **Camera Ready Version Uploaded**
> >
> > Dear AE,
> >
> > Thank you again for all your help. We have now uploaded the camera-ready version along with a video explaining the problem being studied and the gradOL algorithm.
> >
> > Thanks and Regards,
> > Authors

---

> > > ### Author Response · Authors · 2026-01-13
> > >
> > > Dear AE,
> > >
> > > Thank you again for your help in coordinating the reviews and for the positive decision to accept our paper. Could you please let us know if there are any next steps we need to complete?
> > >
> > > Regards,
> > > Authors

---

> > > > ### Comment · Action_Editor_VbFK · 2026-01-13
> > > >
> > > > Dear authors,
> > > >
> > > > No more steps are required. I've verified the camera ready revision, and sorry for the delay on my side.
> > > >
> > > > Best regards,
> > > > AE